# The Adaptation of Maslow’s Hierarchy of Needs to the Hierarchy of Dogs’ Needs Using a Consensus Building Approach

**DOI:** 10.3390/ani13162620

**Published:** 2023-08-14

**Authors:** Karen E. Griffin, Saskia S. Arndt, Claudia M. Vinke

**Affiliations:** 1Department of Population Health Sciences, Division Animals in Science & Society, Animal Behaviour, Faculty of Veterinary Medicine, Utrecht University, 3584 CM Utrecht, The Netherlands; s.s.arndt@uu.nl (S.S.A.); c.m.vinke@uu.nl (C.M.V.); 2The Dog Rehoming Project, Irvine, CA 92604, USA

**Keywords:** anthropomorphism, dog quality of life, dog welfare, dogs’ needs, hierarchy of needs

## Abstract

**Simple Summary:**

The ability to recognize and meet dogs’ needs is central to providing them with good quality of life and overall welfare. This ability is particularly important for dogs that live in shelters and rescue organizations. Unfortunately, humans often struggle to accurately recognize dogs’ needs because they tend to confuse them with their own needs. To combat that tendency, this study sought to develop an objective framework of dogs’ needs. A well-established model of human needs (Maslow’s Hierarchy of Needs) was borrowed from the social sciences and adapted to be relevant to dogs for this purpose. This adaption process involved identifying 37 specific dog needs from the scientific literature and grouping those needs into seven need groups. Those need groups were organized into a hierarchy with levels based on their importance. The adapted hierarchy of needs and the original hierarchy were sent to a group of canine science experts. They were asked to assess several aspects of the adapted hierarchy in comparison to the original and give their feedback, largely to ensure that the former adequately represented the scope of dogs’ needs. After three rounds of feedback, the group of experts had reached as close to a consensus as possible on all aspects of the adapted hierarchy except for items pertaining to the prioritization of any need groups for specific dog categories (e.g., senior dogs). The next phase of this project involves the development of a tool to assess dogs’ quality of life before and after they are adopted from shelters and rescue organizations. During that phase, issues of whether any need groups need to be prioritized for specific dog categories will be readdressed.

**Abstract:**

It is crucial for good dog welfare that humans be able to accurately and adequately recognize and meet dogs’ needs. However, humans may do a poor job of recognizing dogs’ needs due to their tendency to anthropomorphize. The aim of this study was to develop a framework for dogs’ needs that would help humans to recognize and meet their actual needs, thereby improving dogs’ quality of life. Using the Delphi method, to reach as close to a consensus as possible from an expert panel, Maslow’s Hierarchy of Needs was adapted to become the Dogs’ Hierarchy of Needs. To do so, the relevant scientific literature was reviewed to identify 37 specific dog needs, which were group into seven need groups. Those groups were then organized onto five need levels, which were as analogous to Maslow’s Hierarchy of Needs as possible. The expert panel was asked to assess various aspects of the adapted hierarchy in comparison to the original, including face validity, whether they agreed with how the need levels were ordered in terms of importance/priority, and whether they felt that any need groups should be more or less prioritized for any specific dog categories (e.g., senior dogs). After three rounds of expert feedback, there was sufficient consensus for all aspects except items pertaining to the prioritization of any *need groups* for specific dog categories. That aspect of the adapted hierarchy will need to be readdressed in the next phase of this project: the development of a tool to assess the quality of life of dogs that reside in shelters/rescue organizations and post-adoption once they have been rehomed.

## 1. Introduction

To ensure that domestic dogs (*Canis lupus familiaris*) have good quality of life (QoL), humans’ ability to sufficiently and appropriately meet their needs is imperative. However, to fully meet dogs’ needs, first accurately recognizing them, at both species and individual levels, is crucial. Although dog caregivers and owners may have the best intentions for giving dogs good QoL, it is necessary to investigate whether such intentions translate into reality (i.e., if objectively assessed, whether dogs have high QoL). It is possible that caregivers and owners are unaware of the varying levels and types of dogs’ needs (i.e., those at a species level versus those at an individual level) and thus are incorrectly assuming that they are providing their dogs with a higher QoL than they actually are. Ensuring dogs have good quality of life is particularly important in the case of sheltering and rehoming. In this realm, humans may be so focused on “doing good” and “saving lives” that they may be unwilling or unable to recognize dogs’ actual needs; thus, their intentions might have unintended negative impacts on dog welfare.

Humans’ ability to accurately recognize dogs’ needs is often encumbered by their tendency to anthropomorphize [1,2,3]. In doing so, humans project their own motivations, emotional states, and needs onto their dogs and, in turn, “personify” them. Doing so can also lead to misinterpreting a dog’s behaviour, potentially causing them to misunderstand a dog’s needs and emotional state [4]. All of these actions may result in erroneously making the assumption that what is good and beneficial for them as a human is equally so for their dog [1]. Anthropomorphism in this context is differentiated from “critical anthropomorphism”, which can be referred to as the cross-section between empathy and scientific knowledge of animal behaviour and can thus be useful and productive in understanding animals’ experiences and ways of being in the world [4,5]. However, if humans are able to overcome their tendency to anthropomorphize, they will be able to more accurately recognize dogs’ needs by understanding that they may well be different from their own needs. It is then increasingly likely that owners will be more able and willing to meet their dog’s needs, thereby improving their dog’s QoL. Arndt et al. [6] similarly highlighted the positive impact on animal welfare that can occur when animal caregivers and owners are accurately informed of animals’ needs and behaviours and that knowledge is then practically applied. To achieve this goal and to overcome humans’ tendency to anthropomorphize, a framework for conceptualizing dogs’ needs is vital. The aim of this study was to create such a framework by borrowing a well-established model from the human social sciences and adapting it to be relevant to dogs’ needs. For this purpose, Maslow’s Hierarchy of Needs was used, which is a framework of organizing human needs and, thus, human motivations. The five need levels of Maslow’s Hierarchy of Needs are: *Physiological Needs*, *Safety Needs*, *Belongingness and Love Needs*, *Esteem Needs,* and *Self-actualization Needs*. As described by Maslow [7], “Man is a perpetually wanting animal.” Consequently, he argued that “…the appearance of one need usually rests on the prior satisfaction of another, more pre-potent need.” By adapting this conceptualization of ordering and understanding human needs, we hypothesize that it would force us, as humans, to take a step back and shift from merely anthropomorphizing dogs to critically anthropomorphizing them. In doing so, dogs’ qualify of life could be more accurately assessed and ultimately improved. Furthermore, a central goal underpinning this study was to develop a theoretical model of dogs’ needs that would be relevant to numerous and varied populations of dogs (e.g., free roaming dogs), to aid humans in decreasing their potential biases in objectively assessing the QoL of dogs across a wide range of contexts. The concept of dog welfare (or the welfare of any species) and what constitutes *good* welfare is multi-faceted, and there are various approaches to measuring or assessing welfare that have been developed in previous research. Furthermore, other studies have introduced the QoL concept as an assessment tool for animal welfare, which focusses on how the individual animal perceives its own welfare [6,8]. The current study follows this conceptualization of animal welfare and applies it to dogs. The Dynamic Animal Welfare Concept (DAWCon) [6] combines and builds on concepts such as the Five Domains [8,9], the Five Freedoms [10], and quality of life [8]. An in-depth discussion of these concepts is outside the scope of this paper, but we adopt the DAWCon since this concept “assumes the viewpoint of the individual animal and defines welfare as a state that the animal perceives as positive and that evokes positive emotions” [6].

## 2. Materials and Methods

### 2.1. Development of the Initial Theoretical Model

To begin the process of developing the Hierarchy of Dogs’ Needs, an initial list of their needs was compiled by the study authors. The relevant scientific literature was then searched using either the exact wording of the listed needs or related words and terms. For example, when searching the literature for the need for “grooming”, a search for more specific terms that would fall under the umbrella of “grooming” was also performed, such as for “nail clipping” and “haircutting”. This process was performed to increase the likelihood that pertinent literature was not missed, especially since there may be variations in terminology between countries and over time. Whilst searching the literature for the initial list of dogs’ needs, additional needs were realized, which were added to the list. The list was reviewed and revised several times, and subsequent literature searches were performed as revisions were made. Thus, the process of compiling the list was non-linear, in the interest of creating as thorough and exhaustive of list of needs as possible.

The relevant literature that was searched comprised peer-reviewed papers published in scientific journals, in scholarly published books, and other such publications, which included those on topics such as canine behaviour, evolution, and veterinary medicine. Also included was literature pertaining to the needs of other kept animals, such as farm animals, e.g., [10]. All reviewed literature was written in English, and publications from all time periods were included (i.e., no specific publication dates were used as search parameters). Restrictions were not placed on publication years because findings from older studies can still provide insight dogs’ needs, such as those performed with samples of dogs kept in laboratories, especially since studies of this nature would likely not be replicable more recently due to ethical considerations.

The needs were organized in a Microsoft Excel spreadsheet to create a list of *specific needs*. Similar or related needs were grouped together to create *need groups*, which were then organized into need levels. The need levels in the adapted hierarchy were named based on the nature of the need groups contained in them (see Table 1).

The list of needs was then mapped onto Maslow’s Hierarchy of Needs (i.e., the original hierarchy); the nature of the levels in the adapted hierarchy were as analogous to Maslow’s Hierarchy as possible. To do so, the five need levels (i.e., the name of each level) from the original hierarchy (e.g., Psychological Needs) were listed in a column in the spreadsheet following the order of the hierarchy, starting with the bottom (first) level. The need levels were numbered from one to five, with one being the bottom level of the hierarchy and five being the top level. They were numbered for ease of comprehension in the next phase, as discussed later in this section. In the next column of the spreadsheet were need level descriptions for the original hierarchy (i.e., a brief summary of what the need level refers to). In the third column, the need levels for the adapted hierarchy (of dogs’ needs) were listed, again starting with the bottom level. In the fourth column, need level descriptions for the adapted hierarchy were created and listed, which help to clarify that which the need levels in the adapted hierarchy referred. In the fifth column, one of three necessity rankings was assigned to each need level: *mandatory*, *preferred*, and *ideal*. The *mandatory* ranking was qualified as “an acceptable quality of life”, meaning that the specific needs in these levels must be met for a dog to have a quality of life that is acceptable but nothing beyond those needs. The *preferred* ranking was qualified as “a good quality of life”, meaning that, if the needs in these levels are met (in addition to the need levels with a *mandatory* ranking), the dog has a quality of life that exceeds acceptable. The *ideal* ranking was qualified as “the best quality of life”, meaning that, if the needs in these levels are met (in addition to the need levels with *mandatory* and *preferred* rankings), the dog has the best quality of life. In the next column, the need groups for the adapted hierarchy were listed; they were arranged based on the need level to which they belonged. In the seventh and final column of the spreadsheet, the specific needs that belonged to each need group were listed. Maslow’s Hierarchy of Needs did not include need groups or specific needs. However, it was determined that, to ensure that the adapted hierarchy was as robust as possible, it was necessary to capture dogs’ needs at both the species and individual levels. Accordingly, need groups and specific needs were included in the adapted hierarchy.

### 2.2. Expert Panel Recruitment

The adapted and original hierarchies were then sent to a panel of experts to assess the face validity of the adapted version (compared to the original), as well as to provide expert feedback on other aspects of the adapted hierarchy. A list of potential expert panel members was created and comprised people the authors knew personally who had expertise relevant to the study. In the interest of increasing their willingness to participate, it was determined that the panel should comprise people with whom one of more of the authors had a professional relationship. Experts were contacted via email for panel recruitment. They were informed of the purpose of the study and were provided with a brief overview of what would be asked of them as part of the panel. The panel (*n* = 14) comprised people from six countries: the United States, the United Kingdom, the Netherlands, Australia, Belgium, and Chile. By coincidence, all the experts were women. Although exact ages of the experts were unknown, all had been working in their respective fields for a minimum of 10 years, with some experts exceeding 20 years in their fields. The professions/expert qualifications of the expert panel were canine scientists, veterinary behaviourists, and people with long-term, senior roles in animal shelters. Thus, the expert panel comprised individuals with academic (research and teaching) and practical backgrounds. Experts with varied professions and backgrounds were intentionally sought with the goal of receiving robust and dynamic feedback from the panel.

The expert panel was sent the Excel spreadsheet (as described in Section 2.1), an image of Maslow’s Hierarchy of Needs, an image of the Hierarchy of Dogs’ Needs, and instructions for completion via email. Using the Delphi method, which is an iterative process, face validity was established using three rounds of feedback from the expert panel in to reach as close to a consensus as possible. 

### 2.3. The Delphi Method Procedure

In brief, the Delphi method uses the collective judgements or feedback from a panel of experts for the purpose of predicting future events [42,43]. The method does not require that the expert panel comprise a specific or minimum number of members; it was developed on the premise that “two heads are better than one” [42]. When used in research on similar subject matter to the current study, the range of the number of experts involved has been wide. For example, in a study by Griffin [44], four experts were used, whereas a study by Stavisky et al. [45] had an initial panel of 48 experts. Since its development, the Delphi method has been used in a wide range fields of study, such as animal welfare, information systems, and human healthcare to achieve this goal, e.g., [46,47,48,49]. It is an iterative process of feedback gathering to reach as close to an expert consensus as possible. The entire expert panel was contacted for all rounds of feedback, even if they did not participate in the previous round. In the current study, the expert panel was asked to assess three primary components of the adapted hierarchy (in comparison with the original):Whether they agreed with where (i.e., on which level) each need group was placed on the adapted hierarchy;Whether they agreed with how the need levels were ordered in the adapted hierarchy in terms of importance/priority; andWhether they agreed with the labels given to each level.

In addition to these three primary components that the expert panel members were requested to assess, they were asked other questions, such as whether they agreed with how the need levels were categorized in terms of necessity to QoL (i.e., mandatory, preferred, ideal). A full list of the 17 questions that the expert panel were asked can be found in Appendix A.

For the first round of feedback, the panel members were asked to submit their responses to the 17 questions (items) on the spreadsheet they were sent and asked to return the completed spreadsheet via email. For each round of feedback, the panel was emailed an updated spreadsheet, with revisions made based on the feedback from the previous round. For items on which a consensus had been reached in a previous round, they were only asked to address the outstanding items. When the revised spreadsheet was sent for the second and third rounds of feedback, the experts were also sent a list of the changes that had been made (based on feedback from the previous round). For items for which a consensus had not yet been reached, they were given the anonymous feedback from the other panel members to illustrate why a consensus had not yet been reached, as well as to determine whether, based on other experts’ feedback, they wanted to amend their own feedback, such as by gaining new insight that they had perhaps not yet considered.

It was determined that a consensus (or as close to a consensus as possible) had been reached for each aspect of the hierarchy that the panel was asked to assess when there was general agreement from the panel members on that aspect. However, agreement did not necessarily have to be unanimous within the panel, such as if there was agreement among all but one of the experts and that expert’s feedback and opinions were vastly different from the rest of the experts. The number of rounds of feedback was not predetermined at the outset of the study. It was instead dictated by how long it took to reach a consensus on all of the aspects of the hierarchy that the panel was asked to assess or until it seemed unlikely that a consensus was going to be reached on any remaining aspects.

This project was reviewed and approved by the relevant body at Utrecht University, in accordance with Dutch and EU regulations pertaining to animal experiments. This study did not involve animal participants. (See the Institutional Review Board Statement for additional information.)

## 3. Results

Thirty-seven specific needs were identified from the literature, which were organized into 15 need groups. The need groups comprised between one and seven specific needs. Each need level contained between one and 10 need groups, with “Safety Needs” having the most need groups and specific needs. The need levels in the adapted hierarchy were named “Physiological Needs”, “Safety Needs”, “Social Needs”, “Movement Needs”, and “Cognitive Needs”. See Table 1 and Figure 1 for the initial version of the adapted hierarchy that was sent to the expert panel with the questions described briefly in Section 2 and fully in Appendix A. All 15 need groups and 37 specific needs are listed in Table 1. A discussion of the availability and volume of literature that provided an evidence base for the inclusion of the need groups and the specific needs in the hierarchy is provided in the following section.

### 3.1. First Round of Expert Panel Feedback

Responses from the expert panel were received from nine of the 14 panel members (64% of the panel) for the first round of feedback. Not all respondents answered all of the questions asked of them about the hierarchies. Expert panel members provided an abundance of feedback on which need levels several of the need groups were located. This feedback included some disagreement over where need levels several of the need groups were located. Accordingly, with the advice to move multiple need groups from a higher level of the hierarchy to the bottom level (i.e., “Physiological Needs”), four *need groups* were moved to that level with their corresponding specific needs. Those need groups were: ‘provision of physical exercise’, ‘access to shelter/housing’, veterinary care for the treatment of diseases illnesses, injuries, and wounds’, and ‘provision of a place for undisturbed rest’. Because ‘provision of physical exercise’ was moved to “Physiological Needs”, the nature of the fourth level of the hierarchy changed completely and no longer pertained to movement and exercise. Following expert panel feedback, the fourth level was renamed “Integrity Needs” and instead pertained to positive reinforcement based training and behavioural support.

Another aspect of the hierarchy that received considerable feedback, much of it conflicting, pertained to the necessity rankings of each need level as it relates to dogs’ QoL. In the initial adapted version, there were three rankings (i.e., *mandatory*, *preferred*, and *ideal*) (see the Methods section, Table 1 and Figure 1). Whilst the majority of the respondents agreed with all necessity rankings (56% of respondents), three respondents disagreed with one or more ranking. Most notably, one expert panel member felt that all need levels should be ranked as mandatory. Changes to the rankings were made and out to the expert panel for the second round of feedback. The *ideal* ranking was removed entirely, leaving only two rankings—*mandatory* and *preferred*.

### 3.2. Second Round of Expert Panel Feedback

Responses from the expert panel were received from four of the 14 panel members (29% of the panel) for the second round of feedback. After the second round of feedback, as close as possible to a consensus had been reached on 11 of the 17 initial items that were addressed. The remaining six items were sent back to the expert panel for a third round of feedback. Those items were:⚬Do you think that, for any specific dog categories (e.g., seniors, breed/breed types), any of the need groups should be more or less prioritized?⚬If yes, please rate on a scale of −3 to +3 how much any need groups should be more or less prioritized for specific dog categories.⚬If you think any need groups should be more or less prioritized, please state the specific dog categories to which this decision would apply (e.g., seniors, breed/breed type).⚬If you think any need groups should be more or less prioritized for any specific dog categories, please provide references (as a justification for this opinion).⚬Do you feel that, overall, the adapted hierarchy has face validity (compared to the original)?⚬If you have any additional feedback or comments, please add them here.

For the rating scale of the prioritization of any need groups for specific dog categories, less priority was indicated by a rating of −3, −2, or −1, with −3 being the greatest degree of less prioritization. More priority was indicated by a rating of +1, +2, or +3, with +3 being the greatest degree of more prioritization. For example, a senior dog with arthritis might be given a rating of “+3” for the need group, “provision of a place for undisturbed rest”, as rest might be particularly important for this dog in comparison to the general population.

Other than those six remaining items, minimal changes needed to be made to other aspects of the hierarchy based on the feedback received during this round. The changes that were made were small and pertained to the rewording of some need groups and specific needs. For example, the need group “veterinary care for the treatment of diseases, illnesses, injuries, and wounds” was amended to include “the provision of end-of-life care”.

### 3.3. Third Round of Expert Panel Feedback

Responses from the expert panel were received from two of the 14 panel members (14% of the panel) for the third and final rounds of feedback. Of the six items to be addressed in this round, the two panel respondents both agreed that the adapted hierarchy had face validity compared to the original and they did not have any additional feedback or comments. Regarding the other four items, after the three rounds of feedback, a consensus could not be reached for the items relating to whether any of the need groups should be more or less prioritized for specific dog categories (e.g., seniors, breeds/breed types). It was determined that further progress towards reaching a consensus on those items was highly unlikely due to the disparate feedback on this group of items. The final version of the Hierarchy of Dogs’ Needs contains 42 specific needs that were organized into 16 need groups. See Table 2 and Figure 2 for the final version of the hierarchy; changes to it over the course of the adaptation process (compared to the initial adaption) based on feedback from the expert panel—need levels, need groups, and specific needs—have been noted.

## 4. Discussion

Maslow’s Hierarchy of Needs is a pillar of a framework for understanding human needs and motivations used in psychology. It has been adapted by other canine science researchers to conceptualize dogs’ motivations [58]. Because dogs evolved to cohabitate with humans approximately 20,000 or more years ago and adapted to live in the domestic human niche [59,60,61,62,63], adapting a well-established framework of human needs can provide a robust understanding of how varied and complex dogs’ needs are.

Although there have been previous adaptions of Maslow’s Hierarchy of Needs to be relevant to dogs, they have tended to focus on other topics, such as behaviour and dog training, and those adaptations may also offer a more simplified adaptation than the one that the current study sought to develop. Additionally, other adaptations are possibly more geared towards pet/owned dogs, rather than to a range of different dog populations [58,64]. To fully appreciate the breadth of dogs’ needs at both the species and individual levels, it is imperative to provide a highly detailed framework for their needs. That framework in this study strove to expand upon Maslow’s Hierarchy of Needs in its level of detail and specificity, with the goal of mitigating subjectivity in understanding and interpreting dogs’ needs, which is crucial to ensuring dogs have a good QoL. 

Although many of the need groups and specific needs that comprise the first need level in the adapted hierarchy (“Physiological Needs”) are likely unsurprising and obvious, it is still striking how large and diverse the need groups and specific needs are. This strikingness is also true for the second level of the hierarchy (“Safety Needs”), but in this case, some of the need groups and specific needs are less obvious. That the composition of these need levels is so striking suggests that humans are unaware of or underappreciate the complexities of these areas of dogs’ needs.

As previously noted, the process of compiling an initial list of dogs’ needs and reviewing the literature was a non-linear and iterative process, in part due to variations in the volume of published literature on some need groups and specific needs versus others. For example, there was an abundance of literature spanning several decades that provided evidence for the ‘provision of social contact and support’, the ‘provision of cognitive stimulation’, and all of the specific needs contained within those need groups. Conversely, locating literature to provide evidence for other need groups and specific needs was much more challenging, as was the case for ‘provision of grooming and maintenance’ and ‘access to appropriate places for toileting’. There was also much more limited literature than expected for some of the need groups and specific needs that comprised the “Physiological Needs” and “Safety Needs” levels of the hierarchy. This fact could suggest that specific needs, such as “access to fresh water”, “consistently feasible access to food”, and “consistently feasible access to appropriate places for toileting”, are taken for granted or obvious as dogs’ needs; thus, research attention could be more focused on those topics, such as “consistently provided cognitive stimulation” or “consistently provided social contact with other dogs”. Moreover, a deficit of the literature for some need groups or specific needs does not imply that these topics are less important but rather that less research and literature have been devoted to them.

The Delphi method with a panel of experts was used in the current study, as it has been widely used in numerous fields for its merits and more recently has been used in canine science e.g., [44,45,65]. Because dogs’ needs are many and encompass various areas of canine science, using a diverse panel of experts allowed for the development of a comprehensive theoretical model. Experts in various fields of canine science have a rich knowledge base and strong awareness of the relevant scientific literature, which allows them to thoroughly and objectively assess the complex needs of dogs. Additionally, because the expert panel comprised experts from multiple countries, it helped to address and combat any potential cultural biases.

One possible shortcoming of using this method in this study was the rate of attrition of the expert panel from one round to the next. It is reasonable to expect some level of attrition in using this method of consensus building due to its iterative nature, but in this case, by the third and final round of feedback, responses were received from only two panel members (14% of the total panel and 22% of the number who replied to the first round of feedback). One possible explanation for the decrease in the number of responses received over the course of the process is that the panel was asked to assess several aspects of the adapted hierarchy, which may have been time consuming, especially since the panel was being asked to reply multiple times due to the iterative nature of the method. Having a higher response rate would have been preferable to gain as much expert input over the course of the entire process as possible, and not having such a rate is a limitation of this study. However, in the first round of feedback, which is likely when any feedback is most crucial, responses were received from 64% of the total panel. The rate of attrition that occurred over the course of the rounds of feedback could have caused bias in how the process evolved. For instance, if there had been a higher response rate in the second and third rounds of feedback, it is possible that it would have been more challenging to reach a consensus, as there would have been feedback from more experts to consider. This situation could have resulted in the process needing more than three rounds of feedback (to reach a consensus) or possibly not being able to reach a consensus on additional aspects of the adapted hierarchy. One way to address the issue of potentially high attrition rates when using panels of experts would be to have a larger panel to begin with. Another tactic to decrease the likelihood of high attrition rates would be to ensure that the scope of what is being asked of the panel members is explained to them at the outset when they are being recruited to participate, while also reiterating to them that it is an iterative process, so they will be contacted and asked for their participation multiple times. Doing so will help to ensure that panel members are aware of how much time may be involved in participating so that they can decline to participate if they do not feel willing or able to contribute. Finally, an alternative approach to encourage consistent participation throughout the process would be to offer some sort of financial incentive or compensation for the panel’s efforts and contributions.

Aside from the rate of attrition as a limitation of this study, consideration should be given to the composition of the expert panel itself. Although there was some diversity within the panel in terms of where the experts were from and their specific areas of expertise, there was still a degree of homogeneity, which may have caused bias in how the experts assessed the hierarchies. Additional bias could have been caused by how the experts were recruited. All the experts were known personally by one or more of the study’s authors. An anticipated benefit of this fact was a greater likelihood that the experts would be willing to participate, but a negative aspect of this recruitment method may have again been a lack of diversity in their backgrounds or schools of thought. To mitigate these potential biases within the expert panel, experts from other countries and cultural backgrounds could be recruited, namely from non-Western countries and cultures. Additionally, recruiting experts who were not known personally by the authors might allow for a more representative cross-section of experts in the field.

The only aspect of the adapted hierarchy that could not be resolved was whether any of the need groups should be more or less prioritized for specific dog categories. The feedback from the expert panel for this aspect was much more nuanced than that for other areas, as illustrated by some of the experts’ responses below. One reason for this difference might have been because merely determining what constitutes “specific category” of dogs could be challenging. For example, if all dogs of a certain breed are grouped together as a specific category, then should crossbreeds be included with the purebred dogs? Similarly, there seemed to be debate over whether all dogs who meet the criteria for a specific category (e.g., senior dogs) had similar enough needs to be prioritized in the same manner or whether determining the prioritization of needs should happen for each individual dog. Conversely, other experts felt that no prioritization of needs should happen for any dogs, as was suggested by the third response below.

⚬“I think these are fundamental needs that should be met for all dogs and should be personalized for each dog. While seniors may not need as much active exercise as others and may have sensory or cognitive defects that would affect their safety, that should be taken into account for the individual dog. Likewise, some individual dogs may have more anxiety or behavioural issues, but their overall needs are still important.”⚬“I’d opt for a prioritization made for all dogs (so ‘generally’) with side notes on possible additional needs for specific characteristics, such as based on hereditary background (selection for morphological or behavioural characteristics), age/life stage, etc. This may take away the issue of the variation between dogs for such characteristics (as well as the different opinions of the experts), whilst at the same time addressing the importance of attending to the characteristics and/or analysing any (additional or lesser) need levels in a particular dog.”⚬“I think all dogs should be treated equally, or it will get messy, as dogs sexually/socially mature and age at different times. And where do the cross breeds fit into this?”

It is perhaps unsurprising that, with relative ease, a consensus could be reached for what dogs needs are, but determining whether or how those needs should be prioritized on a more individual level becomes stickier due to dogs being such a morphologically and phenotypically diverse species. Nonetheless, because a consensus could not be reached for the questions regarding whether any of the need groups should be more or less prioritized for specific dog categories, this area will be readdressed and investigated once the Hierarchy of Dogs’ Needs is translated into an assessment tool, and data are collected with it in future research. Once that work is done, adjustments will be made to the Hierarchy as needed, based on any meaningful results from future research. We acknowledge that this aspect of the Hierarchy is especially important and requires further research attention.

Although this model of dogs’ needs was developed to be used in the context of dogs residing in shelters/rescue organizations and recently rehomed dogs to assess their QoL in both contexts, its theoretical basis (i.e., quality of life is a function of how well their needs are met) means that it is also applicable to other dog populations. Furthermore, this hierarchy focuses on dogs’ needs themselves and *not* how they are met, so it provides a comprehensive framework for the needs of dogs living in a wide range of environments, such as community-owned dogs, working dogs, laboratory dogs, or stray dogs. 

Because the current study is part of a larger project, the Hierarchy of Dogs’ Needs will now be used to develop a tool to objectively assess the QoL of dogs in a rehoming context pre-adoption and post-adoption as a next step. As discussed in the Introduction, the limited amount of research that has assessed or investigated dog welfare pre-adoption has focused on dogs residing in shelters or kennel environments, e.g., [12]. However, a large proportion of dogs who live in shelters and rescue organizations do not reside in these sorts of environments whilst awaiting rehoming, so it is imperative to ensure they have good welfare and, thus, good QoL, regardless of the environment in which they are living. Many rescue organizations are entirely foster-based, and often shelters or organizations that have a physical building will still have some dogs residing in foster homes. Therefore, it is vital to consider the QoL of dogs residing in any environment. As such, the assessment tool based on this hierarchy will focus on whether their needs are met, rather than specifically how they are met. For example, the *provision of cognitive stimulation* can be achieved through myriad means, so whether or not cognitive stimulation is provided is what is in question and not *how* it is provided, as long as it is in line with the specific needs in this need group.

As noted in the Introduction, the framework of dogs’ needs developed in the current study should help dog owners and caregivers to accurately recognize dogs’ needs, specifically because it is based on the scientific literature and its development, including the opinions and input of experts. This ability, in turn, will allow them to better meet dogs’ needs. We propose that these processes will, most importantly, improve dogs’ QoL and overall welfare in myriad contexts and living environments.

## 5. Conclusions

A comprehensive understanding of the breadth and scope of dogs’ needs will allow humans to more accurately meet dogs’ needs. These processes will improve dogs’ QoL and overall welfare. Though this goal is important for all dog populations, it is particularly important in the context of dogs that reside in animal shelters/rescue organizations and newly rehomed dogs. Maslow’s Hierarchy of Needs provides a useful framework for organizing dogs’ myriad and varied needs, at both the individual and species levels. By adapting this hierarchy to be relevant to dogs’ needs using a consensus building approach with expert feedback, a comprehensive framework of dogs’ needs has been developed. This framework will ultimately lead to improved QoL for dogs, by aiding humans in shifting away from their tendency to problematically anthropomorphize dogs and instead recognize that dogs’ and humans’ needs, wants, and emotions are not necessarily the same as their own.

## Figures and Tables

**Figure 1 animals-13-02620-f001:**
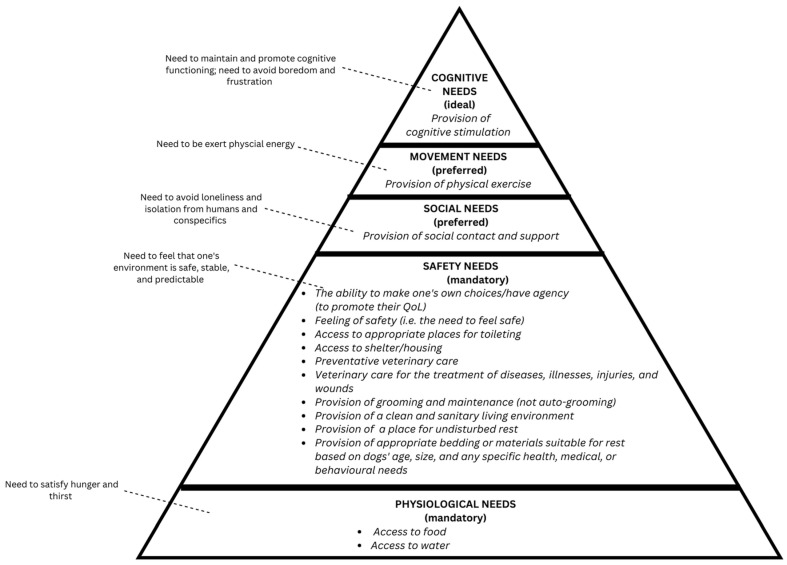
Visual representation of the initial version of the Hierarchy of Dogs’ Needs: need levels, need level descriptions, necessity rankings, and need groups.

**Figure 2 animals-13-02620-f002:**
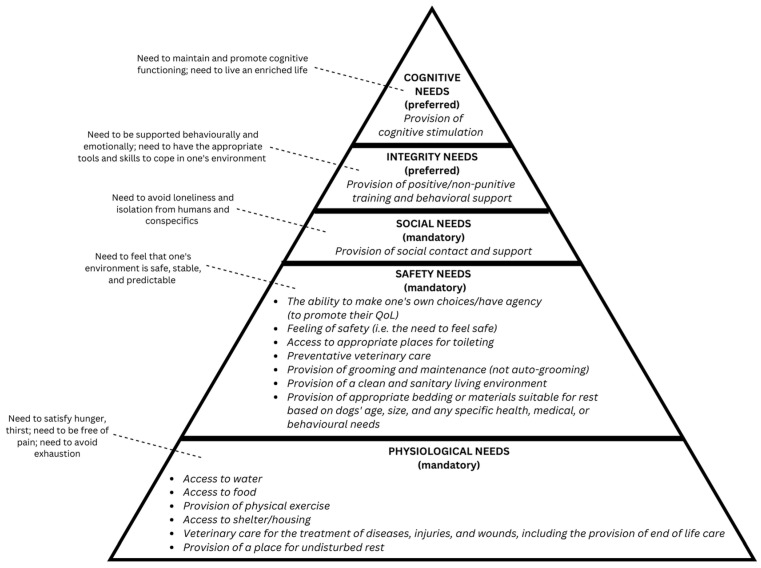
Visual representation of the Hierarchy of Dogs’ Needs (final version): need levels, need level descriptions, necessity rankings, and need groups.

**Table 1 animals-13-02620-t001:** Initial version of the structure of the Hierarchy of Dogs’ Needs (i.e., the adapted hierarchy) in comparison to the original (Maslow’s Hierarchy of Needs): need levels, need groups, and *specific needs*.

Maslow’s Hierarchy of Needs (Original Hierarchy)		Hierarchy of Dogs’ Needs (Adapted Hierarchy)				
Need Level Name	Need Level Description [11]	Need Level Name	Need Level Description	Necessity Ranking of Meeting Each Need Level (as They Relate to QoL)	Need Group	Specific Need
Physiological Needs	*Need to satisfy hunger and thirst*	Physiological Needs	*Need to satisfy hunger and thirst*	mandatory	Access to water	Consistently feasible access [10,12,13,14]
						Fresh water [10,14]
					Access to food	Consistently feasible access [10,12,14]
						Appropriate amounts of food based on dogs’ age, weight, and individual health needs [10,14]
Safety Needs	*Need to feel that the world is organized and predictable; need to feel safe, secure, and stable*	Safety Needs	*Need to feel that one’s environment is safe, stable, and predictable*	mandatory	The ability to make one’s own choices/exercise choice (to promote their own QoL)	Predictability [15,16]
						Controllability [10,15,16,17,18]
					Feeling of safety (i.e., the need to feel safe)	Consistent feeling of safety [10]
					Access to appropriate places for toileting	Consistently feasible access [19]
						Places for toileting that are safe, sanitary, and do not cause any pain or discomfort to the dog [18,19]
					Access to shelter/housing	Consistently feasible access [19]
						Shelter that provides protection from the outside environment, including, but not limited to, snow, rain, temps > degrees, temps < degrees, and is free from draft/drought [10,12,14]
						Shelter/housing meets minimum size specifications based on the size of the dog [14]
					Preventative veterinary care	Routinely executed, as dictated by a veterinarian [10,12,14]
						Care based on dogs’ age, living environment, and individual health needs, including, but not limited to, vaccinations, preventative parasite control, and monitoring of organ functions [10,14]
Safety Needs (continued)	*Need to feel that the world is organized and predictable; need to feel safe, secure, and stable*	Safety Needs (continued)	*Need to feel that one’s environment is safe, stable, and predictable*	mandatory	Veterinary care for the treatment of diseases, illnesses, injuries, and wounds	Care immediately following the occurrence of the wound or injury, or upon noticing signs or symptoms of disease or illness [10,12,14]
						Follow-up care for the treatment of diseases, illnesses, injuries, and wounds, as dictated by a veterinarian [10,12,14]
					Provision of grooming and maintenance (not auto-grooming)	Conducted routinely, including, but not limited to, baths, haircuts, brushing of fur, teeth brushing, nail trimming or filing, and ear cleaning, as dictated by dogs’ age, morphological characteristics, and specific health/medical needs [20]
					Provision of a clean and sanitary living environment	An indoor living environment that is consistently both visibly and invisibly clean and sanitary [18,21,22]
						An outdoor living environment that is consistently both visibly and invisibly clean and sanitary [18,21]
					Provision of a place for undisturbed rest	Consistently feasible access [16,23,24]
						A place that is consistently quiet and calm [16,23,24]
					Provision of appropriate bedding or materials suitable for rest based on dogs’ age, size, and any specific health, medical, or behavioural needs	Consistent access [10,12,14]
						Bedding or materials for rest are clean and dry [10,12,14]
Belongingness and Love Needs	*Need to love and be loved, to belong and be accepted; need to avoid loneliness and alienation*	Social Needs	*Need to avoid loneliness and isolation from humans and conspecifics*	preferred	Provision of social contact and support	Consistently provided social contact with humans [25,26,27,28,29,30]
						Type of social contact with humans that is appropriate for the dog based on their age and behavioural characteristics, is diverse (including humans with varying physical characteristics), and is safe for the dog [20,26,31]
						Duration of social contact with humans that is appropriate for the dog based on their age and behavioural characteristics [20,26,32]
						Consistently provided social contact with other dogs [10,12,14]
						Housing with other dogs when appropriate, as dictated by the dog’s age, breed, health/medical status, and behavioural characteristics [33,34,35]
						Type of social contact with other dogs that is appropriate for the dog based on their age and behavioural characteristics, is diverse (including various types of dogs), and is safe for the dog [10,12,14]
						Duration of social contact with other dogs that is appropriate for the dog based on their age and behavioural characteristics [10,12,14]
Esteem Needs	*Need for self-esteem, achievement, competence, and independence; need for recognition and respect from others*	Movement Needs	*Need to exert physical energy*	preferred	Provision of physical exercise	Consistently provided physical exercise [12]
						Type of exercise that is appropriate for the dog based on their age, health/medical condition, and breed/breed type [12]
						Duration of exercise that is appropriate for the dog based on their age, health/medical condition, and breed/breed type [12]
Self-actualization Needs	*Need to live up to one’s fullest and unique potential*	Cognitive Needs	*Need to maintain and promote cognitive functioning; need to avoid boredom and frustration*	ideal	Provision of cognitive stimulation	Consistently provided cognitive stimulation [35,36,37,38]
						Variability (vs. habituation) [35,39,40]
						Type of cognitive stimulation that is appropriate for the dog based on their age, breed/breed type, health/medical condition, and behavioural characteristics [17,36,37,38,41]
						Duration of cognitive stimulation that is appropriate for the dog based on their age, breed/breed type, health/medical condition, and behavioural characteristics [17,36,37,38]

**Table 2 animals-13-02620-t002:** Final version of the structure of the Hierarchy of Dogs’ Needs: need levels, need level descriptions, necessity rankings, and need groups.

Maslow’s Hierarchy of Needs (Original Hierarchy)			Hierarchy of Dogs’ Needs (Adapted Hierarchy)			
Need Level Name	Need Level Description [11]	Need Level Name	Need Level Description	Necessity Ranking of Meeting Each Need Level (as They Relate to QoL)	Need Group	Specific Need
Physiological Needs	*Need to satisfy hunger and thirst*	Physiological Needs	*Need to satisfy hunger, thirst; need to be free of pain; need to avoid exhaustion*	mandatory	Access to water	Consistently feasible access [10,12,13,14]
						Fresh water [10,14]
					Access to food	Consistently feasible access [10,12,14]
						Food should be palatable and provided in a manner that allows comfort in eating and satiety ^1,2^ [39,50]
						Appropriate amounts of sufficient quality food that is composed of an adequate and balance of macro and micronutrients based on dogs’ age, weight, and individual health needs^1^ [10,14,51]
					Provision of physical exercise ^1^	Consistently provided physical exercise [12]
						Type of exercise that is appropriate for the dog based on their age, health/medical condition, and breed/breed type [12]
						Duration of exercise that is appropriate for the dog based on their age, health/medical condition, and breed/breed type [12]
					Access to shelter/housing ^1^	Consistently feasible access [10]
						Shelter that provides protection from the outside environment, including, but not limited to, snow, rain, temps > degrees, temps < degrees, and is free from draft/drought [10,12,14]
						Access to daylight and fresh air on a daily basis ^1^ [52]
						Shelter/housing is appropriately sized for the size of the dog, ensuring that the dog has adequate room for species specific behaviours (including, but not limited to, lying down, stretching, walking) ^1^ [14]
Physiological Needs (continued)	*Need to satisfy hunger and thirst*	Physiological Needs (continued)	*Need to satisfy hunger, thirst; need to be free of pain; need to avoid exhaustion*	mandatory	Veterinary care for the treatment of diseases, illnesses, injuries, and wounds, including the provision of end-of-life care ^1,2^	Care immediately following the occurrence of the wound or injury, or upon noticing signs or symptoms of disease or illness [10,12,14]
						Follow-up care for the treatment of diseases, illnesses, injuries, and wounds as dictated by a veterinarian^1^ [10,12,14]
					Provision of a place for undisturbed rest ^1^	Consistently feasible access [16,23,24]
						A place that is consistently quiet and calm) [16,23,24]
Safety Needs	*Need to feel that the world is organized and predictable; need to feel safe, secure, and stable*	Safety Needs	*Need to feel that one’s environment is safe, stable, and predictable*	mandatory	The ability to make one’s own choices/have agency (to promote their own QoL) ^1^	Predictability [15,16]
						Controllability [10,15,16,17,18]
					Feeling of safety (i.e., the need to feel safe)	Consistent feeling of safety [10]
					Access to appropriate places for toileting	Consistently feasible access [19]
						Places for toileting that are safe, sanitary, are recognisable to as a toilet area to the dog, and do not cause any pain or discomfort to the dog ^1^ [18,19]
					Preventative veterinary care	Routinely executed, as dictated by a veterinarian [10,12,14]
						Care based on dogs’ age, living environment, and individual health needs, including, but not limited to, vaccinations, preventative parasite control, and monitoring of organ functions [10,14]
					Provision of grooming and maintenance (not auto-grooming)	Conducted routinely, including, but not limited to, baths, haircuts, brushing of fur, teeth brushing, nail trimming or filing, and ear cleaning, as dictated by dogs’ age, morphological characteristics, and specific health/medical needs [20]
					Provision of a clean and sanitary living environment	An indoor living environment that is consistently both visibly and invisibly clean and sanitary [18,21,22]
						An outdoor living environment that is consistently both visibly and invisibly clean and sanitary [18,22]
					Provision of appropriate bedding or materials suitable for rest based on dogs’ age, size, and any specific health, medical, or behavioural needs	Consistent access to appropriate bedding or materials [10,12,14,53]
						Bedding or materials for rest are clean, dry, and safe (for dogs that may chew or ingest bedding) ^1^ [10,12,14,53]
Belongingness and Love Needs	*Need to love and be loved, to belong and be accepted; need to avoid loneliness and alienation*	Social Needs	*Need to avoid loneliness and isolation from humans and conspecifics*	mandatory^1^	Provision of social contact and support	Consistently provided social contact with humans [25,26,27,28,29]
						Type of social contact with humans that is appropriate for the dog based on their age and behavioural characteristics, is diverse (including humans with varying physical characteristics), is safe for the dog [20,26,31]
						Duration of social contact with humans that is appropriate for the dog based on their age and behavioural characteristics [20,26,32]
						Consistently provided social contact with other dogs [10,12,14]
						Housing with other dogs when appropriate, as dictated by the dog’s age, breed, health/medical status, and behavioural characteristics [17,33,34]
						Type of social contact with other dogs that is appropriate for the dog based on their age and behavioural characteristics, is diverse (including various types of dogs), and is safe for the dog; dogs must show willingness for conspecific contact and should not be forced into any such situation ^2^ [10,12,14]
						Duration of social contact with other dogs that is appropriate for the dog based on their age and behavioural characteristics [10,12,14]
Esteem Needs	*Need for self-esteem, achievement, competence, and independence; need for recognition and respect from others*	Integrity Needs^1^	*Need to be supported behaviourally and emotionally; need to have the appropriate tools and skills to cope in one’s environment* ^1^	preferred^1^	Provision of positive/non-punitive training and behavioural support^1^	Behavioural support provided by or overseen by appropriately qualified people (i.e., clinical animal behaviourists, veterinary behaviourists, certified dog trainers) ^1,2^ [54,55]
						Consistently provided behavioural support as needed to address specific behavioural and emotional needs^1^ [54,55]
						Type of behavioural support that is appropriate for the dog based on their age, breed/breed type, and health/medical condition ^1^ [54,56]
Self-actualization Needs	*Need to live up to one’s fullest and unique potential*	Cognitive Needs	*Need to maintain and promote cognitive functioning; need to avoid boredom and frustration*	Preferred ^1^	Provision of cognitive stimulation	Consistently provided cognitive stimulation, including learning new skills ^1^ [17]
						Variability (vs. habituation) [35,40]
						Type of cognitive stimulation that is appropriate for the dog based on their age, breed/breed type, health/medical condition, and behavioural characteristics [57]
						Duration of cognitive stimulation that is appropriate for the dog based on their age, breed/breed type, health/medical condition, and behavioural characteristics [57]

^1^ Revised from original adaptation (see Table 1) based on input from the expert panel during the first round of feedback. ^2^ Revised from original adaptation or second iteration based on the second round of feedback.

## Data Availability

All data are available by contacting the corresponding author, Karen E. Griffin, via email: k.e.griffin@uu.nl.

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
