# Peer review of "The Adaptation of Maslow’s Hierarchy of Needs to the Hierarchy of Dogs’ Needs Using a Consensus Building Approach"

_animals, 2023, doi:10.3390/ani13162620_

Round 1
Reviewer 1 Report
REV-1: The adaptation of Maslow’s Hierarchy of Needs to the Hierarchy of Dogs’ Needs using a consensus building approach
The aim of this study is to adapt the Maslow’s Hierarchy of Needs to the Hierarchy of Dogs’ Needs using a consensus building approach (Delphi method) in order to develop a framework of dogs' needs which would help humans to recognize dog's needs imporving quality of life consequently. This approach sounds very interesting and it could be a useful tool to address the quality of life of shelter dog before and after the adoption. Some work is needed to imporve the manuscript, particularly materials and Methods as well as the discussion needs to be revised accurately.
LL 104-110 it could be better move this part in the discussion
LL 117-122 it is not totally clear how the relevant literature was searched. Did the authors use a review approach using specific terms? If yes, what were the criteria of inclusion/exclusion (i.e., English, peer-reviewed, etc.)
LL 158 give a brief explanation on modified Delphi method
LL 160-161 the authors listed the expertise of experts not the countries. This information was reported in line 163
About the Delphi procedure it is not clear how the consensus was established or calculated by authors across the three rounds. Moreover, it is not specified if the number of rounds was defined a priori or based on the achievement of the consensus. It is useful add more information on the work flow and organization of the Delphi process built by the authors.
About experts’ recruitment, did the authors invite the experts via email? Did the authors use an online system for the administration of the rounds?
LL 229 explain better the scoring: -3 +3
Mat and Met need to be revised in order to explain better the Delphi process. It could be better organize this part as done in the following papers:
Rioja‐Lang, F., Bacon, H., Connor, M., & Dwyer, C. M. (2019). Rabbit welfare: determining priority welfare issues for pet rabbits using a modified Delphi method. Veterinary Record Open,6(1), e000363.
Berteselli, G. V., Messori, S., Arena, L., Smith, L., Dalla Villa, P., & de Massis, F. (2022). Using a Delphi method to estimate the relevance of indicators for the assessment of shelter dog welfare. Animal Welfare,31(3), 341-353.
Subdivide the part in more sub-parties
· Delphi procedure
· Expert panel
· Questionnaire design
· Etc.
LL 337-340 explain possible reasons why the consensus was not reached for the prioritization considering specific dog categories. The discussion should be focused on the results about experts’ prioritization of the needs and opinion on need groups.
Moreover, due to the high attrition rate, it is important point out the limitations of this study.
Author Response
Thank you for your thoughtful and useful feedback. We have addressed all the issues you highlighted in your review. Please see below for our notes and responses for each point raised.
LL 104-110 it could be better move this part in the discussion
This paragraph has been moved to the Discussion section.
LL 117-122 it is not totally clear how the relevant literature was searched. Did the authors use a review approach using specific terms? If yes, what were the criteria of inclusion/exclusion (i.e., English, peer-reviewed, etc.)
Additional detail was added to this part of the Methods section, including more information about how the literature was searched and the criteria used for the literature review.
LL 158 give a brief explanation on modified Delphi method
This was a typo from early work on this study – it was erroneously left in the manuscript. “Modified” has now been removed from the text. A standard Delphi method was used (in comparison to other studies that have used a modified one, such as with non-anonymous experts comprising the panel).
LL 160-161 the authors listed the expertise of experts not the countries. This information was reported in line 163
These sentences have been revised to make better sense.
About the Delphi procedure it is not clear how the consensus was established or calculated by authors across the three rounds. Moreover, it is not specified if the number of rounds was defined a priori or based on the achievement of the consensus. It is useful add more information on the work flow and organization of the Delphi process built by the authors.
A considerable amount more detail has been added to explain the steps of the Delphi method in the context of this study, including how the number of rounds of feedback were determined and how it was determined that a consensus had been reached for each aspect of the hierarchy.
About experts’ recruitment, did the authors invite the experts via email? Did the authors use an online system for the administration of the rounds?
Additional detail has been added about expert recruitment and how their feedback was submitted.
LL 229 explain better the scoring: -3 +3
A more detailed explanation of the rating scale has been added.
Mat and Met need to be revised in order to explain better the Delphi process. It could be better organize this part as done in the following papers:
Rioja‐Lang, F., Bacon, H., Connor, M., & Dwyer, C. M. (2019). Rabbit welfare: determining priority welfare issues for pet rabbits using a modified Delphi method. Veterinary Record Open,6(1), e000363.
Berteselli, G. V., Messori, S., Arena, L., Smith, L., Dalla Villa, P., & de Massis, F. (2022). Using a Delphi method to estimate the relevance of indicators for the assessment of shelter dog welfare. Animal Welfare,31(3), 341-353.
This section was revised to add a considerable amount of more detail on how the Delphi method was applied to this study. The organization of the section was also adjusted to make it easier to follow.
Subdivide the part in more sub-parties
- Delphi procedure
- Expert panel
- Questionnaire design
- Etc.
The Materials and Methods section has been divided into three subsections to better organize and explain the various steps of the study.
LL 337-340 explain possible reasons why the consensus was not reached for the prioritization considering specific dog categories. The discussion should be focused on the results about experts’ prioritization of the needs and opinion on need groups.
This section of the Discussion section has been greatly expanded upon, including by adding quoted responses from some expert panel members to illustrate why a consensus could not be reached for this area of the adapted hierarchy.
Moreover, due to the high attrition rate, it is important point out the limitations of this study.
This has now been specifically noted as a limitation of this study.
Reviewer 2 Report
“The adaptation of Maslow’s Hierarchy of Needs to the Hierarchy of Dogs’ Needs using a consensus building approach
Overall comments
Thank for you the opportunity to review this paper. I found this concept an interesting and relevant one in relation to our application of methods and evidence to ensure maximal welfare of animals in human care. This is a particularly relevant topic in terms of our understanding of animal needs and ow best to convey such information clearly to managers and caregivers of animals.
The premise is clear and the undertaking of the work suitable (please see specific comments) and overall, I feel the work is well written and presented, although a key point would relate to the presentation of the tables that I felt were split in such as way as to lose clarity of contents and made cross-referencing quite tricky, especially in cases of executive dysfunction or more generally for quick and easy reference and accessibility.
I appreciate this work is a first step in a progressive project and would be interested to see the progression of this work more generally relating to the management of the domestic dog.
Simple Summary
No comment – clear, concise, precise.
Abstract
Clear and concise. Appropriate for study. Highlights key areas or relevance including findings/outputs
Introduction
Clear with good definitions and clarity of background of study including scene setting.
Well written and to my knowledge, incorporates relevant key studies and information that is pertinent to this work.
Line 47 – consider amending ‘pet dogs’ to companion or domestic, esp considering point made in line 108
Line 60 – incombered? Spelling? – do you mean encumbered OR incumbered
Line 62 – consider removing/amending use of ‘etc’ for clarity and style
Line 100 – FAWC – reference missing? Should this be Farm Animal Welfare Council?
Materials and Methods
Ethical review noted at end of section – lacks approval number or further details however – these are always useful for cross referencing and audit purposes (line 189)
Line 115-124 – I feel more detail is needed here to evidence literature used – dates, key words/search terms, databases used/accessed as otherwise there could be a perceived/actual risk of missing recent/current data/outputs and/or relying on older sources. Clarity and enhanced detail are suggested
Line 132 – consider replacing ‘the methods’ with ‘this’
Line 154 – expert details is given later but again, I feel more clarity on the selection protocol for choosing experts would be useful – demographic detail, experience, time in role, academic/practical etc – otherwise potential bias – could also be relevant for later note about engagement or lack thereof – was recruiting opportunistic or other?
Results
Line 191 - I’d like more detail on the results/data collection from the literature for clarity and to evidence the base of information used.
Line 202 – ‘feedback on and in disagreement with’ – I feel this needs rewriting for enhanced clarity of meaning
Line 222 – more detail on levels of attrition at each stage of review? Maybe a table or specific detail of numbers
Line 256 – title Figures feels unstructured and they are more tables than figures – applies to 1a-c an 3a-c – I also feel these need combined to a single table for each (before and after) as otherwise, headings are missing and clarity is missing OR ensure headings are carried to each table for full clarity – if single large tables are not acceptable, they need separated out for clarity, perhaps a table for each need level name?
Discussion
Relevant and consistent discussion of findings.
Line 294 – refs? (Some sources increasingly suggest longer than 20, 000 years so perhaps consider ‘more than 20, 000 years)
Line 309 – more detail and discussion about expert panel including discussion of potential biases and experiential/cultural learnings too -massively important in considering HAI and awareness of what would be considered essential and desirable needs for dogs.
Line 323-335 – I’d like more assessment of expert panel selection and potential risk of bias as a result of attrition rates and that those engaged might be more likely to develop the process in a specific way
Line 337-340 – hugely important point to consider, especially as certain breeds and types might be considered to have specific needs and requirements
Conclusion
Clear. Good review and reflection and awareness of relevance of study outcomes to wider canine world.
References
I have not exhaustively gone through these, but all appear fine – present and correct although I have not proofread or cross referenced to check validity of use or correct formatting.
Author Response
Thank you for your thoughtful and thorough feedback. We appreciated that you found this study to be valuable to dog caregivers. That was a key goal of this study. We have addressed all of the issues you highlighted. Our notes and responses to each point are below.
Overall comments
Thank for you the opportunity to review this paper. I found this concept an interesting and relevant one in relation to our application of methods and evidence to ensure maximal welfare of animals in human care. This is a particularly relevant topic in terms of our understanding of animal needs and ow best to convey such information clearly to managers and caregivers of animals.
The premise is clear and the undertaking of the work suitable (please see specific comments) and overall, I feel the work is well written and presented, although a key point would relate to the presentation of the tables that I felt were split in such as way as to lose clarity of contents and made cross-referencing quite tricky, especially in cases of executive dysfunction or more generally for quick and easy reference and accessibility.
These are very large tables so splitting them into separate tables, while still making them easily comprehensible and accessible, is tricky. However, they been redivided in a manner that should make them easier to follow and cross reference. The newly divided tables have replicated column headings and rows are no longer split between pages.
I appreciate this work is a first step in a progressive project and would be interested to see the progression of this work more generally relating to the management of the domestic dog.
Simple Summary
No comment – clear, concise, precise.
Abstract
Clear and concise. Appropriate for study. Highlights key areas or relevance including findings/outputs
Introduction
Clear with good definitions and clarity of background of study including scene setting.
Well written and to my knowledge, incorporates relevant key studies and information that is pertinent to this work.
Line 47 – consider amending ‘pet dogs’ to companion or domestic, esp considering point made in line 108
This has been changed to ‘domestic dogs’.
Line 60 – incombered? Spelling? – do you mean encumbered OR incumbered
This was a typo – it has been changed to ‘encumbered’.
Line 62 – consider removing/amending use of ‘etc’ for clarity and style
This has been removed.
Line 100 – FAWC – reference missing? Should this be Farm Animal Welfare Council?
The reference (for Farm Animal Welfare Council) has been added.
Materials and Methods
Ethical review noted at end of section – lacks approval number or further details however – these are always useful for cross referencing and audit purposes (line 189)
As per Dutch regulations, studies that do not involve animal participants do not receive an ethics approval number. However, the declaration of approval for this study is stored in the archives of the Animal Welfare Body, which is who grants ethical approval. This additional information on the policy was added to the Institutional Review Board Statement at the end of the manuscript, and was referenced in this part of the Methods section.
Line 115-124 – I feel more detail is needed here to evidence literature used – dates, key words/search terms, databases used/accessed as otherwise there could be a perceived/actual risk of missing recent/current data/outputs and/or relying on older sources. Clarity and enhanced detail are suggested
Additional detail has been added to this portion of the Methods section, including more information on search terms and parameters use for the literature used.
Line 132 – consider replacing ‘the methods’ with ‘this’
This has been changed.
Line 154 – expert details is given later but again, I feel more clarity on the selection protocol for choosing experts would be useful – demographic detail, experience, time in role, academic/practical etc – otherwise potential bias – could also be relevant for later note about engagement or lack thereof – was recruiting opportunistic or other?
Additional details about the experts have been added here, and a greater discussion of the shortcommings of the expert panel (including recruitment method) has been added to the Discussion section.
Results
Line 191 - I’d like more detail on the results/data collection from the literature for clarity and to evidence the base of information used.
A bit of additional detail was added here, but a more in depth overview of the volume and availability of literature for some needs compared to others was added to the Discussion, as this is where it seemed better suited to go.
Line 202 – ‘feedback on and in disagreement with’ – I feel this needs rewriting for enhanced clarity of meaning
This has been reworded (and broken into two sentences) for clarity.
Line 222 – more detail on levels of attrition at each stage of review? Maybe a table or specific detail of numbers
Details were added to each mention of level of attrition to make in this section to add clarity, as it didn’t seem there was enough information to warrant adding a table. (It would probably be more worthwhile if there were more than three rounds of feedback to note attrition rates for.) The rounds of feedback portion of the Results section was also divided into subsections, which should also make the process and results for each round easier to follow for the reader.
Line 256 – title Figures feels unstructured and they are more tables than figures – applies to 1a-c an 3a-c – I also feel these need combined to a single table for each (before and after) as otherwise, headings are missing and clarity is missing OR ensure headings are carried to each table for full clarity – if single large tables are not acceptable, they need separated out for clarity, perhaps a table for each need level name?
These are very large tables so splitting them into separate tables, while still making them easily comprehensible and accessible, is tricky. However, they been redivided in a manner that should make them easier to follow and provide additional clarity. The newly divided tables have replicated column headings and rows are no longer split between pages. It is understandable that these do look more like tables than figures, but because of the formatting style/template used in this journal for tables isn’t suitable for the material contained in these, it was determined that they should be left in their original format and referred to as figures.
Discussion
Relevant and consistent discussion of findings.
Line 294 – refs? (Some sources increasingly suggest longer than 20, 000 years so perhaps consider ‘more than 20, 000 years)
The wording has been amended to reflect that domestication may have happened longer than 20,000 years ago. References are at the end of the sentence.
Line 309 – more detail and discussion about expert panel including discussion of potential biases and experiential/cultural learnings too -massively important in considering HAI and awareness of what would be considered essential and desirable needs for dogs.
A discussion of the risk of potential biases due to the composition of the expert panel and the recruitment method has been added.
Line 323-335 – I’d like more assessment of expert panel selection and potential risk of bias as a result of attrition rates and that those engaged might be more likely to develop the process in a specific way
A discussion regarding potential biases due to attrition rates has been added.
Line 337-340 – hugely important point to consider, especially as certain breeds and types might be considered to have specific needs and requirements
This has been elaborated on to underscore the importance of doing this.
Conclusion
Clear. Good review and reflection and awareness of relevance of study outcomes to wider canine world.
References
I have not exhaustively gone through these, but all appear fine – present and correct although I have not proofread or cross referenced to check validity of use or correct formatting.
Reviewer 3 Report
The article focuses upon an objective framework of dogs’ needs. The authors applied the Maslow’s Hierarchy of Needs from human social sciences and adapted it to the relevant needs of dogs in order to recognize and establish good quality of life for latter. The focus here was in particular on shelter and rescue dogs. The authors identified specific dog needs from the scientific literature and grouped these into seven need groups. The need groups were organized into a hierarchy with levels based on their importance and sent to a group of canine science experts. The experts were requested to assess the aspects of the hierarchy to ensure that it effectively characterized the scope of dogs’ needs. The aim of the work was to develop a tool to assess dogs’ quality of life before and after they are adopted from shelters and rescue organizations.
The issue is important and the manuscript clear, relevant for the field and presented in a well-structured manner. However the subject matter is not new and has been published a number of times before by different authors (see von Reinhardt C https://www.animal-learn-verlag.de/buecher/hunde/336/stress-bei-hunden Foltin S.,https://www.animal-learn-verlag.de/detail/index/sArticle/962/sCategory/81)
Thus novelty is not given even though in a paper (as compared to books see above) this question may not have been presented before. The results are significant, the conclusions justified and supported. The article is well written and of interest to the reader. There is therefore an overall benefit to publish the articles and thus an advance in current knowledge.
The response rate of the experts was not good but this limitation was adequately deliberated in the discussion section.
There are some cumulative sentences and some unconcise sentences as well as spelling and grammar mistakes. Please correct those – some are already marked in the attached paper.

Author Response
Thank you for your thoughtful and positive feedback on this study. We appreciate that you have found there to be value in this study and that it is deserving of being published. We have addressed all of the issues you highlighted. Our notes and responses to the points you raised are below.
The article focuses upon an objective framework of dogs’ needs. The authors applied the Maslow’s Hierarchy of Needs from human social sciences and adapted it to the relevant needs of dogs in order to recognize and establish good quality of life for latter. The focus here was in particular on shelter and rescue dogs. The authors identified specific dog needs from the scientific literature and grouped these into seven need groups. The need groups were organized into a hierarchy with levels based on their importance and sent to a group of canine science experts. The experts were requested to assess the aspects of the hierarchy to ensure that it effectively characterized the scope of dogs’ needs. The aim of the work was to develop a tool to assess dogs’ quality of life before and after they are adopted from shelters and rescue organizations.
The issue is important and the manuscript clear, relevant for the field and presented in a well-structured manner. However the subject matter is not new and has been published a number of times before by different authors (see von Reinhardt C https://www.animal-learn-verlag.de/buecher/hunde/336/stress-bei-hunden Foltin S.,https://www.animal-learn-verlag.de/detail/index/sArticle/962/sCategory/81)
We do acknowledge that the subject matter is not new, nor is this study the first to adapt Maslow’s Hierarchy of Needs to be relevant to dogs. However, we feel that the novelty of this study is in how a dog’s quality of life is qualified (as a function of how well their needs are met). Similarly, we feel that we have uniquely adapted Maslow’s Hierarchy for a specific purpose, in comparison to other adapted versions, such as those that have been created to illustrate dogs’ training/behavioral needs or their motivations. We also feel that the adapted hierarchy in this study has a level of specificity that versions adapted elsewhere do not have.
Thus novelty is not given even though in a paper (as compared to books see above) this question may not have been presented before. The results are significant, the conclusions justified and supported. The article is well written and of interest to the reader. There is therefore an overall benefit to publish the articles and thus an advance in current knowledge.
The response rate of the experts was not good but this limitation was adequately deliberated in the discussion section.
There are some cumulative sentences and some unconcise sentences as well as spelling and grammar mistakes. Please correct those – some are already marked in the attached paper.
The sentence structures and spelling/grammer mistakes have all been addressed.
Round 2
Reviewer 1 Report
The manuscript is improved and is suitable for publication.
All of my comments and suggestions were considered and addressed.
I don't have any other comments.